# Antarctic glaciers export carbon-stabilised iron(II)-rich particles to the surface Southern Ocean

Rhiannon L. Jones [1,2] ✉, Jon R. Hawkings [3,4], Michael P. Meredith [1], Maeve C. Lohan [2], Oliver W. Moore [5], Robert M. Sherrell [6], Jessica N. Fitzsimmons [6,7], Majid Kazemian [8], Tohru Araki [9], Burkhard Kaulich[8] & Amber L. Annett [1]

Iron is an essential micronutrient for phytoplankton and plays an integral role in the marine carbon cycle. The supply and bioavailability of iron are therefore important modulators of climate over glacial-interglacial cycles. Inputs of iron from the Antarctic continental shelf alleviate iron limitation in the Southern Ocean, driving hotspots of productivity. Glacial meltwater fluxes can deliver high volumes of particulate iron. Here, we show that glacier meltwater provides particles rich in iron(II) to the Antarctic shelf surface ocean. Particulate iron(II) is understood to be more bioavailable to phytoplankton, but less stable in oxic seawater, than iron(III). Using x-ray microscopy, we demonstrate co-occurrence of iron and organic carbon-rich phases, suggesting that organic carbon retards the oxidation of potentially-bioavailable iron(II) in oxic seawater. Accelerating meltwater fluxes may provide an increasingly important source of bioavailable iron(II)-rich particles to the Antarctic surface ocean, with implications for the Southern Ocean carbon pump and ecosystem productivity.

Iron (Fe) is an essential micronutrient for phytoplankton yet is highly insoluble in oxic seawater and therefore scarce in the surface ocean. Iron is the main limiting nutrient in the Southern Ocean, a region critical to global primary productivity and carbon cycling over annual and glacial-interglacial timescales[1]. Inputs of Fe from the Antarctic continental shelf, such as the West Antarctic Peninsula, are strongly linked to enhanced primary productivity both on the outer shelf region, and the largely Fe-limited Antarctic Circumpolar Current[2–6]. Particulate Fe (pFe) is an important component of the bioavailable Fe supply to the global ocean[7–11]. Recent studies using glaciogenic Patagonian dust show that primary Fe(II)-rich silicates are bioavailable and promote a high growth response in Southern Ocean phytoplankton comparable

to that of dissolved Fe, and more so than Fe(III)-rich minerals such as ferrihydrite, highlighting the importance of valence and mineralogy on bioavailability[10,12]. Furthermore, particles dominated by pFe(II)-minerals proximal to an ice stream on the East Antarctic coastline were linked with high productivity in nearby parts of the Southern Ocean[13], and Fe in waters proximal to Antarctic glaciers was found to be directly more available to phytoplankton[14]. No study has investigated the Fe component of glaciogenic particles in the surface seawater of the West Antarctic Peninsula (WAP).

The WAP, defined as the shelf region west of the Antarctic Peninsula (Fig. 1) is home to ~674 glaciers, of which 87% are retreating[15–17] and >98% are marine-terminating. Glaciers produce and

[1]British Antarctic Survey, Cambridge, UK. [2]School of Ocean and Earth Science, University of Southampton, Southampton, UK. [3]Department of Earth and Environmental Science, University of Pennsylvania, Philadelphia, PA, USA. [4]iC3, Department of Geosciences, UiT, The Arctic University of Norway, Tromsø, Norway. [5]Department of Environment and Geography, University of York, York, UK. [6]Departments of Marine and Coastal Sciences and Earth and Planetary Sciences, Rutgers University, New Brunswick, New Jersey, USA. [7]Department of Oceanography, Texas A&M University, College Station, Texas, USA. [8]Diamond Light Source Ltd., Harwell Science & Innovation Campus, Didcot, UK. [9]Institute for Molecular Science, Okazaki, Japan. ✉e-mail: rhines@bas.ac.uk

transport high loads of freshly weathered material to the ocean water column, providing an important direct and indirect source of pFe and dissolved Fe (dFe, operationally defined as <0.2 μm) through subglacial discharge, glacial meltwater, iceberg rafted debris, and deposition and resuspension of glaciogenic shelf sediments[18–23]. Physical weathering of continental Fe(II)-rich silicates mobilises particles with a high Fe(II) content[24], implying high potential bioavailability. However, Fe oxidation kinetics suggest that pFe(II) should oxidize rapidly upon entering the oxygen-rich waters of the Southern Ocean[25], rendering it less bioavailable. We examined the Fe(II)/Fe(III) speciation in individual suspended particles from three glacially-influenced bays along the WAP, and determined whether stabilisation of particulate Fe(II) by organic carbon is likely[26]. We show that glacier-derived pFe is likely becoming increasingly important in biogeochemical cycling on the west Antarctic Peninsula shelf given continued glacial retreat, with implications for Fe and carbon cycling in the coastal Southern Ocean.

## Results and discussion

### WAP glaciogenic particles are Fe(II)-rich

We present an Fe speciation study of suspended particles in WAP surface waters, to examine the role of melting glaciers on the Southern Ocean Fe cycle. Scanning X-ray microscopy (SXM) was used to collect Fe and C X-ray absorption near edge structure (XANES) spectra on 61 individual particles (~0.2 – 2.7 μM), to determine Fe(II)/Fe(III) speciation, and carbon composition and co-location with Fe. Samples were collected from the glacial meltwater-rich surface waters (<1 m depth) of three glaciated bays on the WAP. The bays are situated (from north-to-south) on King George Island, Anvers Island, and Adelaide Island (Fig. 1). Despite differing provenance and associated weathering pathways at the three sites, we find that Fe-rich particles along the WAP coast commonly resemble mixed-valence Fe species, with a mean Fe(II):Fe$_{sum}$ across sites of 41 ± 8 % (1 s.d.), which is considerably higher than the Fe(II):Fe$_{sum}$ of typical dust-derived surface ocean particles of

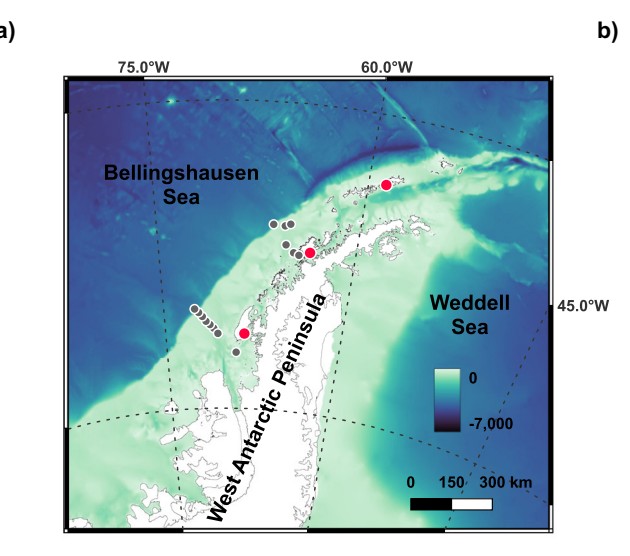

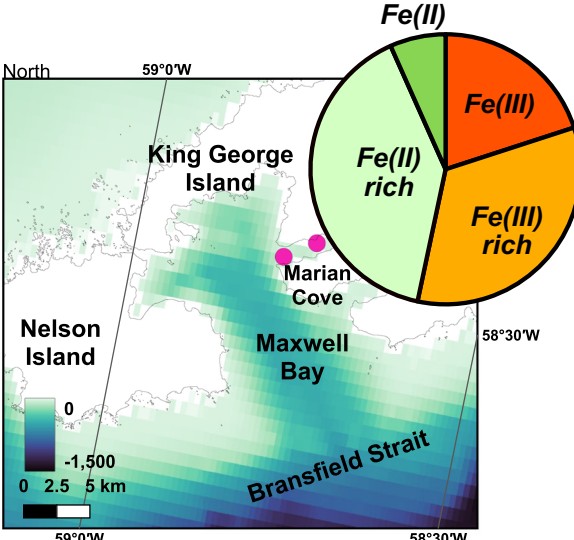

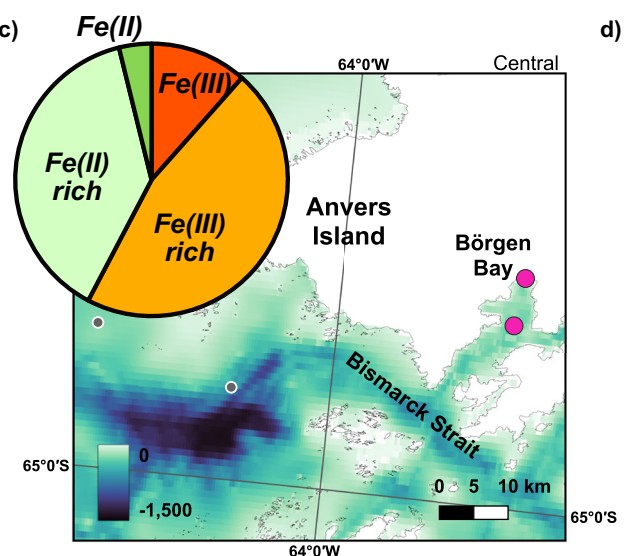

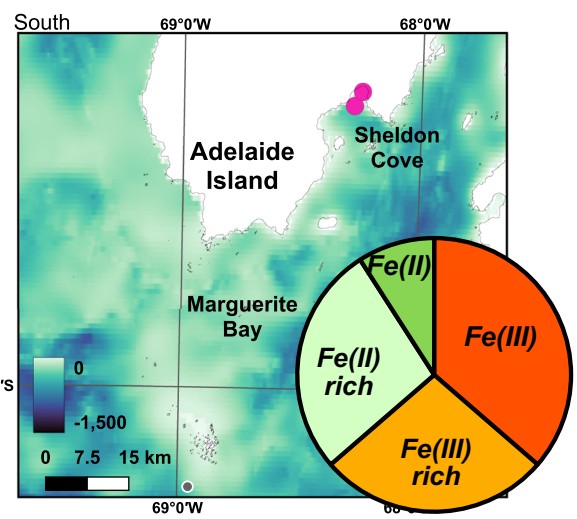

**Fig. 1 | Maps showing the location and bathymetry of the three study sites along the West Antarctic Peninsula.** Panel (**a**) shows the broad West Antarctic region, with the three study sites marked by pink dots and the Palmer Long Term Ecological Research (LTER) lines used in this study marked with dark grey dots. Panels (**b–d**) show the smaller scale regions from north (King George Island), central (Anvers Island) to south (Adelaide Island), with the location of particle collection for X-ray Absorption Near Edge Structure (XANES) marked in pink. The pie charts indicate the Fe(II) and Fe(III) classification of all particles measured at each site. Classification categorisation is described in the methods, following Bourdelle et al.[78]. The maps were created using open-source mapping software QGIS. The bathymetry is ETOPO 2022 at 30 arc-second resolution (NOAA National Centers for Environmental Information. 2022) and the coastline data is from the SCAR Antarctic Digital Database, 2023.

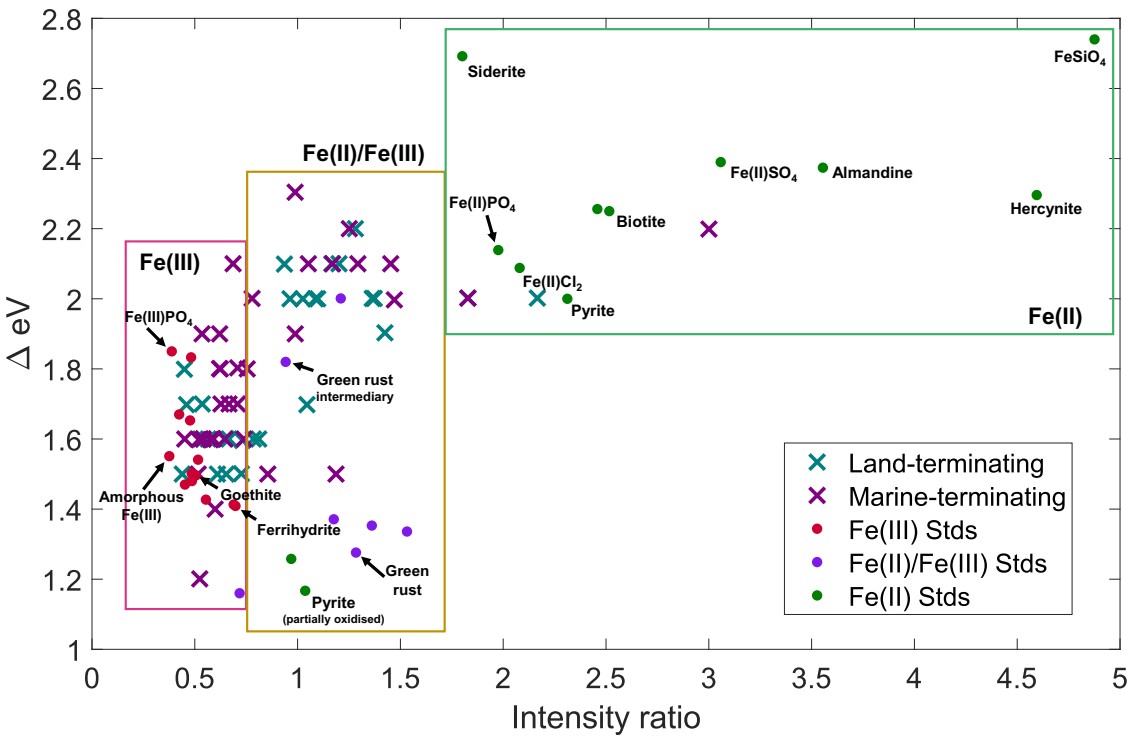

**Fig. 2 | Mapping of the X-ray Absorption Near Edge Structure (XANES) spectra L$_{3-a}$ and L$_{3-b}$ peak differences in energy ($\Delta$eV), and peak intensity ratios for iron-rich particles measured in this study and standards from literature, provided in Supplementary Information Table S2.** All particle spectra analysed are given a data point, and the data is distributed between marine-terminating ($n = 34$) and land-terminating ($n = 27$) systems. Approximate criteria for Fe(III), Fe(II)/Fe(III) mixed, and Fe(II) are given. Adapted from Von der Heyden (2012)[12]. The full dataset along with published standards are given in Supplementary Table S2.

~5%[24]. WAP particles resemble a range of Fe(II)-rich to Fe(III)-rich minerals, as shown by mapping the key Fe L$_3$-edge XANES features of natural particles against mineral standards (Fig. 2). Particles plot adjacent to common Fe(III)-rich oxyhydroxides like ferrihydrite and goethite, green rust (mixed Fe(II)/Fe(III)), and primary Fe(II)-rich minerals such as pyrite and biotite[27].

**Organic carbon stabilizes pFe(II) at the WAP**

The prevalence of Fe(II)-rich minerals consistent with green rust and pyrite in fully oxic waters (surface O$_2$ was 330 – 380 µM) indicates that some stabilisation mechanism must be acting to preserve the Fe(II) in glaciogenic particles across the WAP. Importantly, no significant speciation difference was observed between glacier-proximal ( < 0.5 km) and glacier-distal (3–8 km) stations within the three bays, suggesting that particles are transported at least this distance with no significant oxidation of Fe(II) to Fe(III) (independent two-tailed t-test, $t(60) = 0.33$, $p = 0.74$). Applying the Fe(II) oxidation rate equation from[28], suggests a half-life of uncomplexed Fe(II) of 40 – 50 min, very likely shorter than the timescale of transport between these stations, according to maximum current speeds of <0.4 m s$^{-1}$ measured across the WAP[29,30]. Furthermore, Fe(II):Fe$_{sum}$ is independent of approximated particle size, except for the five largest particles ( > ~1.6 µm), suggesting that preferential loss by sinking of the largest particles would leave most of the Fe(II)-rich particles in suspension.

Using SXM, we examined Fe and C co-location in areas ~5–20 µm$^2$. Where Fe was present (30 imaged areas), C was also present in 63% of areas ($n = 19$) and statistically co-located with Fe ($r = >0.3$) in 50% of areas ($n = 15$) (e.g. Fig. 3, Supplementary Table S1). Hotspots of Fe at the sub-micron scale were often entombed within a carbon-rich matrix (Fig. 3). Previous work has shown that aggregation of Fe with carbon-rich matter slows the oxidation of Fe(II) to Fe(III) in seawater[26,28], and in glacial systems[27], which could potentially preserve Fe(II)-rich particulates until delivery to surface ocean primary producers. In summary, we demonstrate statistically significant co-location of Fe-rich particles entombed within C-rich material in the surface ocean where it may be available to phytoplankton.

To interrogate which C functional groups are associated with the Fe-associated C-rich material, C K-edge near edge X-ray absorption fine structure (NEXAFS) spectra were collected (Supplementary Fig. S1). All C spectra co-located with Fe displayed a characteristic aromatic peak ( ~ 285 eV), varying in relative peak amplitude between samples. Spectra also displayed peaks at higher energies, commonly related to aliphatic (287.1 – 288.2 eV) and carboxylic (287.7 – 289.0 eV) functional groups, and inorganic carbonate ( > 290.2 eV)[31] (Fig. S1). Carboxylic functional groups, in particular, commonly interact with transition metals via carboxylic ligand exchange (predominantly Fe(III)-rich minerals), and are commonplace in the marine dissolved organic matter pool[32–35]. Surface seawater dissolved organic carbon (DOC) concentrations ranged from 40.4 µM (King George Island) to 82.2 µM (Adelaide Island), exceeding the deep water values in modified Circumpolar Deep Water (mCDW) of 39.3 – 56.0 µM. Regional DOC over 40 – 50 µM likely indicates additional labile and semi-labile DOC from microbial sources relative to the refractory background DOC[36,37]. Comparing our NEXAFS C-spectra with published standards[38], enrichment in aromatic, aliphatic and carboxylic functional groups all suggest the presence of protein-like compounds, which are commonly associated with metal-binding ligands, and represent a relatively labile form of C[26]. Labile organic matter-bound Fe can be highly bioavailable[39] and labile forms of organic C undergo remineralisation more readily than recalcitrant forms, providing pathways for

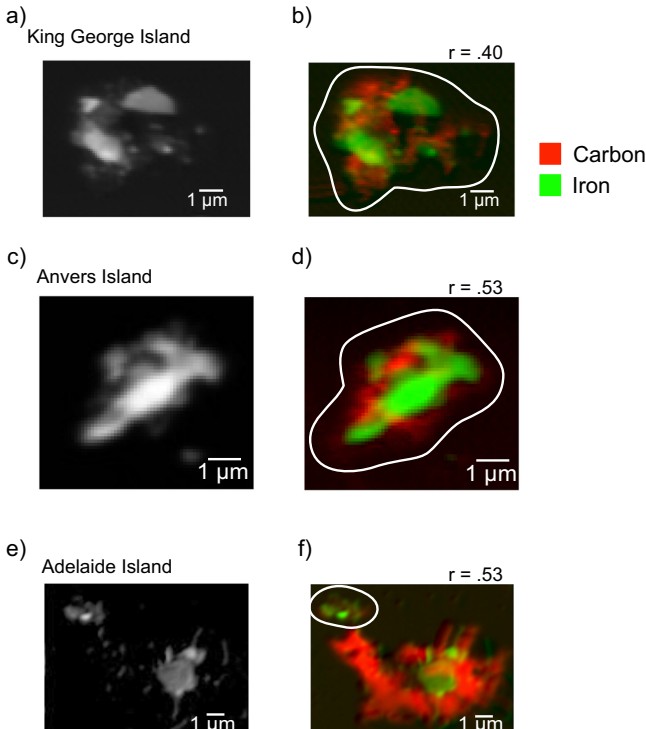

**Fig. 3 | Scanning X-ray Microscopy (SXM) images of regions of interest from each site providing evidence of Fe and C co-location.** The left panel shows the Fe optical difference map, where black indicates no Fe and white indicates the highest optical density of Fe. The right panel provides colourised red-green composite maps of Fe (green) and C (red) SXM images. Brighter colours indicate a higher abundance of Fe and C. (**a**, **b**) are images taken for samples from the northern site (King George Island), (**c**, **d**) the central site (Anvers Island) and (**e**, **f**) the southern site (Adelaide Island). The areas outlined in white in the composite images represent the areas selected for pixel co-localisation analysis in ImageJ using the *coloc2* plugin. The *r* values represent the Pearson's R correlation coefficient corresponding to the pixels within these areas.

**Table 1 | Fe speciation term for particles depending on $\frac{Fe^{3+}}{\sum Fe}$, defined by comparison with standards**

| Dominant Fe speciation | $\frac{Fe^{3+}}{\sum Fe}$ |
|---|---|
| Fe(II) | <20 |
| Fe(II) rich mix | 20 – 50 |
| Fe(III) rich mix | 50 – 80 |
| Fe(III) | 80 – 100 |

subsequent Fe mobilisation for uptake primary producers[40]. However, deconvolution of C functional group peaks is more complex in natural samples than pure functional groups when C is associated with Fe (oxyhydr)oxides; adsorption to ferrihydrite by carboxyl ligand exchange has been found to reduce the amplitude of the carboxyl peak and shift it to lower energies by up to 0.4 eV[31]. Inferring the lability of the observed carbon associated with aliphatic/carboxylic peaks is therefore uncertain, highlighting the need for additional research on carbon bound to Fe-rich minerals from natural marine samples, and consideration of the influence of carbon functional groups upon the bioavailability of the pFe pool, in the context of long-term preservation of organic carbon and Fe[31,41,42].

## Transport of pFe(II) offshore

The potential for pFe to provide bioavailable Fe to Antarctic phytoplankton will somewhat depend on the source and subsequent transport offshore. All sites had a measurable meltwater component, determined from seawater oxygen isotope data and salinity, coincident with metal-rich particulate material[43]. South of 64 °S, glacial retreat rates are high, driven by the basal melting of marine-terminating glaciers by incursions of warm mCDW onto the shelf[46]. On Adelaide Island, there is a reduction in optical transmission at depth that follows the Gade Line, a line of well-characterised slope in potential temperature/salinity space that indicates the presence of glacial meltwater[44] (Fig. S3). At Adelaide Island, this reflects submarine melting of the Sheldon Glacier face that drives meltwater-derived sediment inputs. According to concurrent radium isotope data, intense sediment-bottom water interactions are observed at the glacier grounding line, as well as upward entrainment of sediment and nutrients by the glacial plume[43]. Alongside this, particle influence was higher in the surface waters (<1 m depth) relative to the water column, indicating multiple input pathways for glacial material. Seawater concentrations for pFe were 29–36 nM, and dFe were 3.2–4.4 nM (Supplementary Table S3). These concentrations far exceed Southern ocean averages of <0.4 nM and <0.1 nM for pFe and dFe, respectively[45], and are strong evidence of proximity to an Fe source, such as delivery of sediment-rich meltwater (e.g.[46]).

At Börgen Bay on Anvers Island, evidence also points to multiple meltwater input pathways, including subglacial meltwater discharge, submarine melting of the glacier face, and melt from ice calving. As for Adelaide Island, glacial meltwater at depth, defined using the Gade line, appears to correlate with sediment inputs at the glacier face[44] (Fig. S3). Meteoric water contributed a substantial 3–8% to surface waters (<1 m depth), decreasing with distance from the glacier terminus, as did surface pFe, pMn, pAl and pTi concentrations, with very high pFe ranging from 30 to 143 nM, and dFe ranging from 4.4 to 7.9 nM. Particulate Mn and dissolved Mn averaged 2.5 and 5.4 nM, respectively.

At the land-terminating Fourcade Glacier (King George Island, northern WAP), meltwaters were derived from surface runoff and concentrated within the upper 2 m of the water column, with a mean contribution of 5.8 %. Maximum radium and thorium activities imply surface processes dominate delivery of glacial particulate material at this location[43]. This site had the highest particulate metal concentrations: surface pFe and dFe concentrations ranged from 370 to 580 nM and 3.2 to 6.4 nM. Notably, pMn and dMn at King George Island were very high, measuring on average 14.4 nM and 38.2 nM, respectively. The surface ocean pAl concentrations at the King George Island site (1.4–3.1 μM) were 3–7 times higher than at the two sites further south, where pAl ranges from 0.07 to 0.44 μM, despite a comparable surface meltwater contribution. As a proxy for lithogenic input, and in combination with higher pFe and pMn (Supplementary Table S3), this indicates a higher meltwater-associated total input of material from the land-terminating glacier at King George Island. The ratio of surface pFe:pAl (mol/mol) at King George Island was 0.17–0.20, the lowest of all sites, compared to a crustal ratio of 0.21 and bulk WAP sediment ratio of 0.26[47,48]. Lack of enrichment in pFe relative to pAl[48,49] supports the interpretation that sediment inputs are unaltered surface inputs of lithogenic material, rather than upwelled from seafloor sediments.

The broader mineralogical composition of surface ocean material is investigated using Scanning Electron Microscopy-Electron Dispersive x-ray spectroscopy, which reveals a mineralogy rich in Si and Al, with measurable Na, Ca, Mg and Fe. Notably, the moderate but ubiquitous measurable Si ($\bar{x}$ = 19 ± 1.7 wt%, $n$ = 318, 1 s.d.) and Al ($\bar{x}$ = 8.3 ± 2.1 wt %) content is consistent with the proximal terrestrial rock[50–53] (Supplementary Table S4, Supplementary Data File 1). Intra-site variability in Si, Al and Fe wt % indicates a potential difference in

source rock geology or alteration since mobilisation: particles at King George Island exhibited a total Al wt % of 11.6 ± 5.0 wt % (median, $n = 104$, 1 s.d.), ~1.5–1.75 times higher than the two sites further south, where total Al measured 7.8 ± 3.3 wt % ($n = 88$, 1 s.d.) and 6.6 ± 3.9 wt% ($n = 97$, 1 s.d.), for Adelaide Island and Anvers Island, respectively, relative to an upper crustal average of 7.96 wt%[54]. These geologies are consistent with production of lithogenic particles dominated by Si- and Al- bearing primary aluminosilicate minerals by physical erosion[55], which is associated with a higher Fe(II) content and has demonstrated high bioavailability in controlled incubation experiments[10,24]. In addition, randomised individual particle analysis data identified high Fe (> 20 wt%) in 4.8% ($n = 15$) of particles interrogated, indicative of Fe (oxyhydr)oxides (Fig. S3). These results provide supporting evidence that Fe(II)- and Fe(III)-bearing minerals are present in suspended particulate material, and that particles are likely to have originated from glacial physical erosion.

Together, these findings indicate that high concentrations of mixed Fe(II)/Fe(III) and Fe(II)-rich particles are exported by glacial meltwaters or entrained from sediments into the water column at the glacier terminus, providing a large source of potentially bioavailable pFe(II) to the WAP. The export of these particles offshore depends predominantly upon aggregation rates and sinking, and circulation pathways[56]. Although colloidal and dissolved Fe can have residence times of months to years in the mixed layer[57], Fe is subject to aggregation in coastal environments[58], and this will lead to the loss of glaciogenic particles from surface waters before biological utilisation in the mixed layer. The potential for subsequent resuspension and transport of Fe-rich sediments may provide a previously underappreciated flux of Fe(II)-rich particles to the WAP shelf and the Antarctic Circumpolar Current, but this is difficult to ascertain with our data. It is therefore important to examine the pFe fraction retained in the surface ocean, relative to that subject to removal across the WAP shelf.

Annual meltwater exports along the WAP from King George Island, Anvers Island, and Adelaide Island are estimated at 2, 8, and 10 Gt y$^{-1}$, respectively, although strong seasonal and annual variability in meltwater export is inferred from across-shelf surface $\delta^{18}$O measurements[59,60]. Across the northern Antarctic Peninsula, exceptionally high winds can intensify surface water currents[59], including the north-eastward flowing coastal Bransfield Current. The Bransfield Current can exceed 0.25 m s$^{-1}$ proximal to Maxwell Bay[61], ~10 km from the King George Island study site (Fig. 1). Maxwell Bay also experiences rapid wind-driven transport of large volumes of suspended material[62], and low productivity during summer is attributed to sediment-driven light limitation here[63]. Previous research indicates that Fe-rich material exported from the northern WAP islands, transported by the Bransfield Current to the often Fe-limited Antarctic Circumpolar Current, fuels productivity in the Scotia Sea[2,64]. Our findings suggest that Fe(II)-rich particles are a significant component of this Fe flux.

Meredith et al.[60] provide multi-year evidence that WAP glacial meltwaters persist across the Antarctic shelf in surface waters during summer, and surface meltwater transport offshore has previously been linked with elevated pFe concentrations away from the coast[46]. To evaluate the export of Fe-rich glaciogenic particles to the Antarctic shelf, we compare our results with samples collected across the Palmer-Long Term Ecological Research (LTER) grid during austral summer, 2015. In 2015, surface (< 5 m) meteoric water inputs, pFe and pMn were highest along the coast, decreasing with distance across the shelf (Fig. 4, Fig. S4). The change in pFe with total meteoric water and distance from the coast is best described using a power law function ($r^2 = 0.56$ and 0.43, respectively). The pMn concentration also exhibited a correlation with meteoric water and distance from coast, best fit by linear functions ($r^2 = 0.59$, $r^2 = 0.41$, respectively). These results suggest that both pFe and pMn share a common

meltwater-derived source and are exported offshore, with decoupling of pFe from pMn, likely related to different settling and/or aggregation rates for Fe-bearing minerals. For example, pFe decreased with distance from 2.5 nM, 13 km from the coast, to 0.4 nM 127 km from the coast off Anvers Island. Offshore of Adelaide Island, pFe concentrations decreased from 4.8 to 0.18 nM, between 8 and 160 km from the coastline. Shelf break values are comparable with those measured in previous years, and exceed typical dFe values in the region of <0.1 nM[46]. Although these represent small levels of total retention, here 16% and 4% from Anvers Island and Adelaide Island to the outer shelf, respectively, pFe could nonetheless provide an important Fe source to Fe-limited outer shelf waters. The offshore transport of meltwater enriched in pFe and pMn is notably not in the direction of the along-shelf Antarctic Peninsula Coastal Current[65–67], providing evidence that during austral summer, meltwaters transport pFe offshore to the Antarctic shelf water column. Assuming a conservative offshore transport of 0.01 m s$^{-1}$, meltwater would travel 160 km in 6 months. Using Stokes Law as a first-order approximation of particle settling (see Supplementary Information Note 1), we calculate that for a particle of diameter 0.2 μm – 1 μm, with a density lower limit of biotite (3090 kg m$^{-3}$) and an upper limit of pyrite (5000 kg m$^{-3}$), the settling rate would be 0.4 – 20 m in 6 months. For an aggregate of varying organic matter to mineral ratio, we calculate that over a 6 month period, aggregates up to 5 μm could be retained in the top 40 m. Accordingly, most particles measured in this study would be retained in the surface mixed layer from distances equal to the distance from the WAP coast to the shelf break.

We show evidence that glacial meltwaters sampled across the WAP export high concentrations of potentially bioavailable Fe(II)-rich particles to Antarctic marine waters, and that a significant component of this pFe is aggregated with organic C. Export of pFe by meltwaters beyond the nearshore environment and into the potentially Fe-limited waters of the Southern Ocean is also demonstrated, where the supply of this reactive Fe pool could support primary productivity. The carbon-complexed pFe(II)-rich material in the surface may become directly available for biological uptake, and we speculate that the fraction that sinks from the water column and reaches Antarctic shelf sediments may form an additional important reactive Fe pool, later resuspended or dissolved to form part of a bioavailable flux of Fe to the mixed layer[46,68]. With continued and accelerated retreat of ~600 WAP glaciers[16], the meltwater-Fe flux to the surface Southern Ocean is expected to rise, increasing the relative importance of this mechanism to the global marine Fe cycle.

## Methods
### Sample source regions
Coastal sample collection was carried out on the R.R.S James Clark Ross JR19002 Icebergs 3 cruise (31 Dec. 2019 – 06 Feb. 2020) to the West Antarctic Peninsula (WAP). The cruise sampled from three bays along the WAP characterised by retreating glaciers: Marian Cove, King George Island (62.2 °S, 58.8 °W); Börgen Bay, Anvers Island (64.7 °S, 63.4 °W); and Sheldon Cove, Adelaide Island (67.5 °S, 68.3 °W) (Fig. 1). Fourcade Glacier (King George Island) is a land-terminating glacier (since 2017), whilst Sheldon Glacier (Sheldon Cove) and William Glacier (Börgen Bay) are both marine-terminating[16].

Across-shelf sample collection for particulate metals, salinity, and stable oxygen isotopes, was carried out on the LMG 15-01 Palmer-Long Term Ecological Research (Pal-LTER) Cruise (29 Dec. 2014 – 09 Feb. 2015), aboard the ARSV Laurence M. Gould. The Pal-LTER grid is visited ~annually and covers an area 600 x 200 km perpendicular to the Antarctic Peninsula coast (Fig. 1a), encompassing the waters offshore of two sites (Börgen Bay and Sheldon Cove) sampled by the Icebergs 3 expedition.

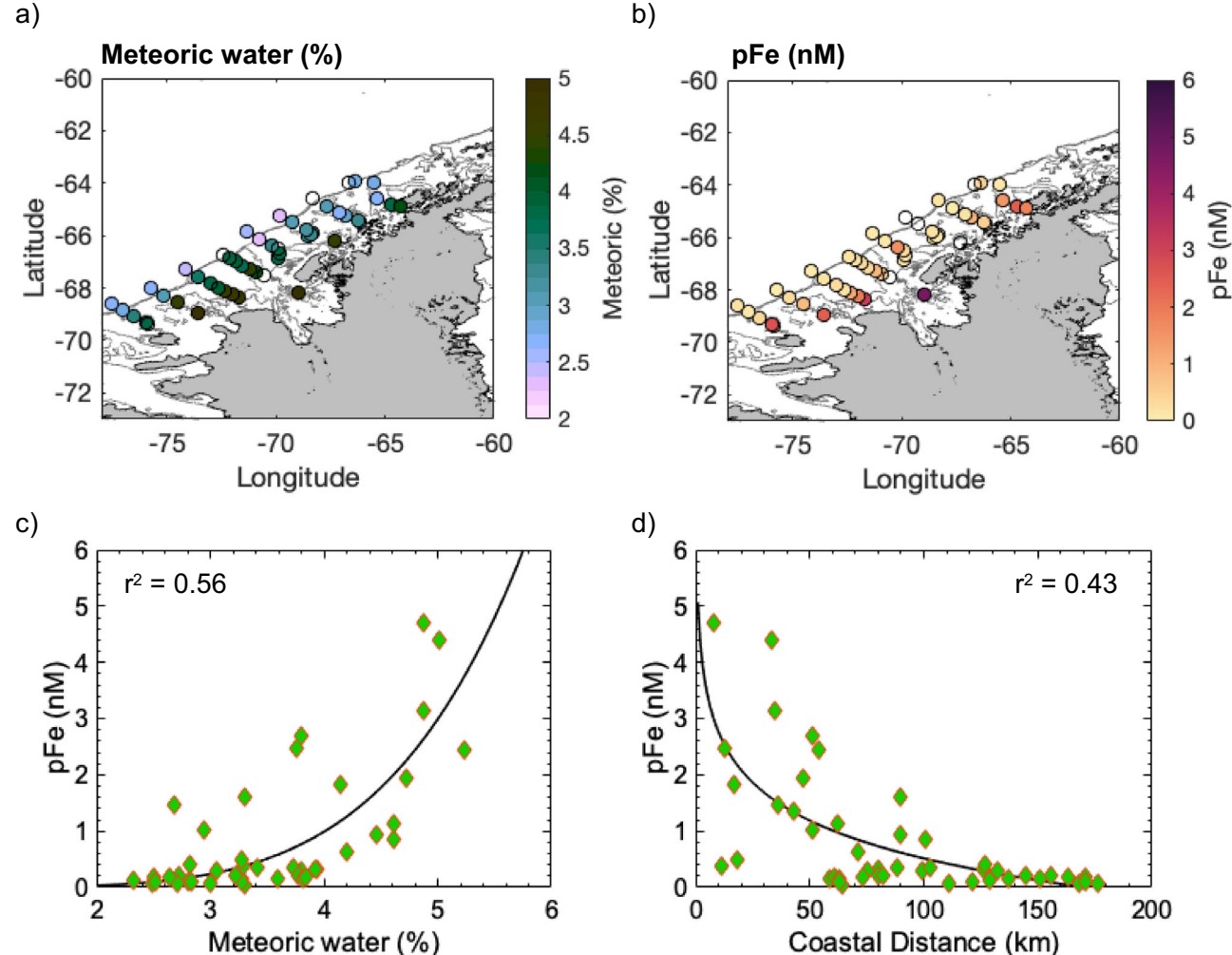

**Fig. 4 | Palmer Long Term Ecological Research (LTER) data from 2015 showing the relationship between meteoric water contribution, distance from coast, and particulate Fe (pFe) measured across the West Antarctic Peninsula.** In (**a**) surface meteoric contribution (%), and (**b**) surface pFe, are plotted onto an area map of the West Antarctic Peninsula. The black contours represent 100 m isobaths, the darkest of which is the 500 m isobath (north-west boundary). In (**c**) pFe concentrations are plotted with meteoric % and fit with the power function $pFe = 0.001x^{4.966}$, ($r^2 = 0.56$) and in (**d**) pFe (nM) is plotted with distance from coast, fitted with the power function $pFe = 102x^{-0.01} - 97.14$ ($r^2 = 0.43$).

## Sample collection

Surface seawater for particulate SXM, total particulate digest, dissolved micronutrients, salinity, and stable oxygen isotopes were collected by hand using zodiac boats during the JR19002 Icebergs 3 cruise (Supplementary Table S5). Acid-cleaned bottles were rinsed and filled with surface seawater over the bow as the boat moved slowly into the wind to avoid contamination from the Zodiac. Bottles were immediately capped and returned to the ship in the dark.

During the LMG15-01 Pal-LTER Cruise, particulate metals were collected from trace metal clean towed-fish and rosette systems. Full details of towfish system[50] and Teflon-coated Niskin-X rosette samplers[54] have been published previously. Briefly, acid-cleaned polyethylene tubing was deployed off the side of the ship to avoid contamination from the hull, and water samples were collected from ~2 m depth while towing at up to 11 kts. Water was pumped on board into a HEPA-filtered lab. For rosette sampling (5 m sample), particles were pressure-filtered directly from Niskin bottles in a HEPA-filtered clean room. For particulate metals, seawater was collected in acid-cleaned 4 L bottles, pressure filtered onto 25 mm Supor® Poly-ethersulfone (PES) filters (0.45 μm pore size) and frozen immediately in clean PetriSlides.

Particles for SXM analysis were collected by filtration onboard immediately after sampling. Using an acid-cleaned (10% HCl) poly-sulfone Nalgene® 47 mm filter unit, acid-cleaned tubing, and vacuum pump, 500 mL of sample was filtered through 47 mm 0.4 μm poly-carbonate (Whatman® Cyclopore®) filters. For total particulate digest, 4 L of seawater was pressure filtered onto 25 mm Supor® (PES) filters (0.45 μm pore size) and frozen immediately. Both PES and Poly-carbonate filters were placed in petri-slides and frozen immediately at −20 °C. Dissolved phase samples were filtered using a 0.2 μm Acropak® cartridge filter and peristaltic pump and collected into acid-washed 125 mL HDPE bottles (dissolved metals), and 250 mL (DOC). All sampling and handling was performed according to GEOTRACES protocols[69].

## Sample analyses

For both sample sets, PES filters (JR19002, $n = 9$; LMG15-01, $n = 54$) were subjected to complete acid digestion using 10% HF/50% $HNO_3$ (ultrapure, distilled or Fisher Optima) at 135 °C for 4 h following GEOTRACES protocol[70]. Digest solution was stored at 4 °C in 3 mL of 0.8 M $HNO_3$. Elemental composition for LMG15-01 particles was quantified by sector-field ICP-MS at Rutgers University (Finnigan

MAT Element 1) analysis. For JR19002 particles, at the University of Southampton (ICP-MS, Finnigan MAT Element 1). Nine multi-element external standard dilutions were created gravimetrically from single-element standards to quantify the range of sample concentrations. Samples were corrected for blank signals using blanks for acids, PTFE vials, and filters. Indium (1 ppb) was used as internal standard to correct for drift and matrix effects. For the LMG15-01 dataset, sample concentrations below the instrument limit of quantification were discarded, which for pFe and pMn was 49.7 pmol kg$^{-1}$ and 3.90 pmol kg$^{-1}$ (10 × blank s.d.), following the approach of[71]. Analytical precision was better than 5%. For the JR19002 dataset, sample concentrations were well above the instrument limit of quantification.

## Dissolved micronutrients

Concentrations for dissolved metals were determined using Sea-FAST preconcentration followed by ICP-MS analysis. In short, 10 ml aliquots of seawater samples were automatically extracted using the commercially available SeaFAST pico system (Elemental Scientific, Inc.) after online buffering to pH -6.5 using ammonium acetate and a 25-fold pre-concentration into 10% (v/v) ultrapure nitric acid (Optima grade, Fisher Scientific). Iron was quantified using isotope dilution, while Mn used external matrix-matched standards. Analytical replicates were run -every sixth sample and were typically 1–3% deviation about the mean. Precision, from repeat measurements of an in-house seawater standard was typically 3% or better, and accuracy was assessed by repeat measurements of consensus standards SAFe S and SAFe D2 Pacific Ocean reference seawater samples as reported in[68,71]. SAFe S gave Fe = 0.102 ± 0.021 ($n$ = 29 over a 2 month period; consensus 0.093 ± 0.008). SAFe D gave Fe = 0.968 ± 0.066 ($n$ = 39 over 1 year; consensus 0.933 ± 0.023). Sample size for the JR19002 dissolved metals was 4 – 5 per site.

## Dissolved organic carbon

Concentrations for DOC (250 mL) were measured with a Shamdzu TOC-L$_{CHN}$, with a high-sensitivity catalyst. Sample concentrations were calibrated using certified standard potassium hydrogen haphthalate (1000 ± 10 mg C L$^{-1}$, Sigma Aldrich). Repeat standard measurements had a variance of <5%. The limit of quantification was 0.039 mg L$^{-1}$ (5 S.D. × limit of baseline).

## Hydrographic properties

Salinity measurements from both Zodiac boat surface samples and CTD rosette full water column are as described in ref. 43 for analysis on a Guildline 8400B salinometer. Salinity is here presented on the Practical Salinity scale. We report CTD profiler potential temperature, light transmission, and salinity. Transmission (%) represents the percentage of incident light that passes through an optical sensor (transmissometer) fixed to the CTD, as an indicator of suspended matter concentration in the water column.

## Stable oxygen isotopes

Samples were collected from the CTD or zodiac boats (JR19002) or from the CTD rosette or underway system (LMG15-01) and stored in 50 ml glass vials sealed with stoppers and crimps. Oxygen isotope samples were analysed -9 months later at the UK's National Environmental Isotope Facility at the British Geological Survey, using the CO$_2$ equilibration method[72] with an Isoprime 100 mass spectrometer plus Aquaprep device. Isotope measurements were calibrated against internal and international standards including VSMOW2 and VSLAP2. Based on duplicate analysis, analytical reproducibility was <0.05 ‰ for all samples.

## Quantifying freshwater inputs

To trace freshwater provenance, we accompanied salinity measurements with stable oxygen isotopes ($\delta^{18}O$), which reflect the magnitude and distribution of freshwater inputs in surface waters. The $\delta^{18}O$ of precipitation becomes lower toward the poles due to preferential evaporation of the lighter isotope and preferential rainout of the heavier isotope. Therefore, $\delta^{18}O$ can be extremely low in glacial ice, providing a useful tracer of glacial discharge into the ocean[73]. Sea ice formation and melt has minimal impact upon $\delta^{18}O$[74]. Concurrent measurements of salinity and $\delta^{18}O$ can therefore separate sea ice melt, meteoric water (precipitation and glacial melt), and ambient seawater, enabling a three-endmember mass balance method to calculate the relative contributions of each water source[74]. Derivation of the chosen endmember values (Supplementary Table S6) is previously described[60,63]. Sensitivity studies find that the uncertainties in the final freshwater fractions are better than 1% for point values[75].

## Spectroscopy analysis

Analysis of the presence and coordination of iron (Fe) and carbon (C) of particles collected onto filters at 6 stations ($n$ = 6) was performed by collecting SXM maps, XANES and NEXAFS using the SXM Beamline (I08) at the UK's Diamond Light Source facility (Harwell Science and Innovation Campus, Oxfordshire) over a 3-day and 5-day period.

## SXM sample preparation

Polycarbonate filter samples were cut into quarters under trace metal-clean conditions. At the Harwell Diamond facility, one quarter was placed in acid-clean microcentrifuge tubes. Particulates were resuspended from the filter using -3 mL of ultra-high purity water using an ultrasonic bath. A small aliquot (<10 µL) of this suspended sample was then pipetted onto an EM Silicon Nitride Membrane grid and allowed to air dry. Each SiN grid was then mounted on the I08 beamline. Sample imaging and mapping started at 250 nm /pixel resolution over 50 × 50 µm$^2$. Within this area 3-8 regions of interest were imaged, first at the C K-edge (280 and 290 eV) and then at the Fe L-edge (705 and 711 eV), performed in this order to avoid unnecessary beam damage of carbonaceous material. Areas for imaging were selected based on beam transmission (between 20 and 80%), and presence of enough background absorbance (i.e. 'white space') to facilitate baseline correction. Measurable Fe was largely ubiquitous, so few maps were discarded from analysis. Pairs of optical difference maps ($n$ = 30) were then analysed for Fe and C spatial co-location using the pixel-based correlation plugin for ImageJ (Fiji), *coloc2*. For each map where Fe and C were present, a region of interest was defined, and each pixel analysed for co-location of both elements to produce a Pearson's coefficient of correlation. A Pearson's coefficient >0.30 is defined as moderate correlation and was observed in 15 out of the 19 maps where Fe and C were both present. Six out of the 19 maps produced a Pearson's coefficient of >0.50. Figure 3 shows examples of Fe and C in aggregates demonstrating robust evidence of co-location at each site. Supplementary Table S1 provides the co-localisation Pearson's coefficient for all samples analysed. For further analysis of particulate Fe and C speciation, new regions of interest (ROI) (4 – 6 µm$^2$) were raster-scanned at a spatial resolution of 30-50 nm/pixel with a dwell time of 2 ms. A stack of images over the full Fe L-edge from 700 eV to 735 eV, and carbon K-edge from 290 to 320 eV were acquired at a spectral resolution of 0.2-1 eV, with higher resolution over the edge energies. Regions were selected for full stacks based on Fe and C presence, determined by quick Fe and C difference maps to maximise available beamtime, as stacks tended to take 2–4 h each and beamtime was limited.

Analyses were conducted at the particle scale for Fe; XANES spectra were collected for each pixel. The Fe L$_3$-edge and L$_2$-edge peaks on XANES spectra contain information on the oxidation state and coordination environment of Fe-rich particles[76]. Images were

processed (dark signal subtraction, stack alignment, and ROI delineation) using Mantis and axis2000 software. Subsequent Fe ROI spectra were normalized in Athena. Using the relative intensity of the $L_{3-a}$ and $L_{3-b}$ peaks of normalised spectra, the proportion of Fe(II) and Fe(III) was calculated using the approximation given in Bourdelle et al.[77]:

$$\frac{Fe^{3+}}{\sum Fe} = \frac{R_{L_3} - 0.1867}{0.01991} \qquad (1)$$

$$R_{L_3} = \frac{I(L_{3-b})}{I(L_{3-a})} \qquad (2)$$

We then define the Fe phases according to the sum of $Fe^{3+}$ to the sum of total Fe $\left(\frac{Fe^{3+}}{\sum Fe}\right)$ as provided in Table 1.

Bourdelle et al. [77], apply this approximation to silicates finding that the $L_{3-b}/L_{3-a}$ intensity ratios correlate linearly with $\left(\frac{Fe^{3+}}{\sum Fe}\right)$ with little scatter ($r^2 = 0.96$). Ferrihydrite is not a silicate, however, applying the same approximation to the standard used gives an $\left(\frac{Fe^{3+}}{\sum Fe}\right)$ of 95%. We therefore assume this equation yields an appropriate estimate, considering the demonstrated relationship between intensity ratio and Fe(II)/Fe(III) for non-silicates (i.e.[13,27]).

Carbon K-edge spectra were also aligned, and regions of interest defined using Mantis and axis2000. MagicPlot was used to normalize carbon spectra and perform peak deconvolution to identify the relative contribution of different carbon functional groups.

### SXM beam damage
Beam damage can induce a valence shift from Fe(III) to Fe(II)[78]. To determine if this occurred during our experiments, a beam sensitivity study was performed on one sample. Beam damage potential was assessed by repeatedly scanning ($n = 5$) an Fe-rich particle collected from the study region in 2 ms increments, to determine the change in the Fe L-edge spectra or intensity ratio with increasing beam exposure. Analysis showed no significant variation in peak ratio between the $L_{3-a}$ and $L_{3-b}$ peaks (Fig. S5). Similar sensitivity experiments by others showed no observable change in valence state over equivalent exposure times[27].

### Scanning electron microscopy
Nine samples (3 from each glaciated bay) were studied using scanning electron microscopy-energy dispersive X-ray spectroscopy (SEM-EDS) to determine the solid phase mineralogy and elemental composition. A total of 289 particles comprises the full dataset, -30 particles from each sample. A small strip of sample filter was removed and transferred onto a conductive and adhesive carbon tape mounted on an aluminium SEM sample holder. Samples were coated with 20 nm of gold and placed on the stage of the microscope under low vacuum conditions. Chemical and morphological properties of the particles were determined by spot analysis and in some cases elemental mapping using an SEM (Carl Zeiss Leo 1450VP SEM) equipped with an Oxford Instruments X-Act $10\,mm^2$ area SEM-Energy Dispersive Spectrometer. The SEM was set to a vacuum $<10^{-4}$ mbar to avoid electrical charging, and analysis was performed at a working distance of $12 - 13$ mm, with an accelerating voltage of 15 kV. Resolution was -1 μm. Carbon (C) and gold (Au) were excluded from elemental mapping due to the use of both elements in the preparation of samples. Example SEM images of particles from each bay are provided in Supplementary Fig. S6, with accompanying EDS spectra. Quantification of element abundance was calibrated against the suite of SEM-EDS AZtec in-built standards. The inbuilt AZtec EDS calibration setup requires routine calibration of the

beam energy and the beam resolution, which was performed using a cobalt metal standard at 20 kV and a working distance of 15 mm.

## Data availability
The XANES/NEXAFS spectra and SEM-EDS elemental composition datasets underlying this manuscript are available on figshare: XANES/ NEXAFS: https://doi.org/10.6084/m9.figshare.28351280.v1. SEM-EDS: https://doi.org/10.6084/m9.figshare.28430921.v1. All ship-based JR19002 CTD data are available through the BODC dataset repository: doi:10.5285/d9633a6c-d27e-30b4-e053-6c86abc07104. Additional datasets from the JR9002 expedition are available through the British Antarctic Survey Polar Data Centre: JR19002 dissolved and particulate metal concentrations: https://doi.org/10.5285/930e4c7e-0415-42db-b98a-d7e0b16c5ab8. JR19002 dissolved organic carbon concentrations: https://doi.org/10.5285/deaec6fa-9654-426d-ba67-756776b25cd7. The LTER-2015 particulate metals dataset is available here: https://doi.org/10.6084/m9.figshare.28351997.v1. All auxiliary LTER datasets are available here: https://doi.org/10.6073/pasta/65c43a4688eccf8ca7cc3a2d07a0cc78.

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

## Acknowledgements

The authors are grateful for the support of the officers, crew, and scientists aboard the JR19002 and LMG15-01 cruises for all assistance and support. We thank, in particular, Dr. Hugh Ducklow, Chief Scientist on LMG15-01, for allowing time and space for the trace element sampling among all the required LTER operations. The study was funded as part of the Radium in Changing Environments: A Novel Tracer of Iron Fluxes at Ocean Margins (RaCE:TraX) grant (NE/P017630/1) (A.L.A.) and the National Environmental Research Council INSPIRE Doctoral Training Partnership (NE/S0072101) (R.L.J.). We acknowledge Diamond Light Source for time on I08 beamline supported by the STFC (UKRI) under proposal MG30572 and MG32502 (A.L.A., R.L.J., J.R.H., M.C.L., O.W.M.). Finally, we wish to thank all three reviewers for their insight and feedback, which greatly strengthened the article.

## Author contributions

R.L.J., A.L.A., M.C.L., J.R.H., M.P.M., O.W.M., R.M.S. and J.N.F. collected and analysed samples. R.L.J., A.L.A., M.C.L., J.R.H., M.P.M., O.W.M. conceived and designed the Fe and C speciation and mapping work. R.L.J., A.L.A., M.C.L., J.R.H., O.W.M., conducted the spectroscopic experiments with support from M.K., T.A. and B.K. R.L.J. performed all data analysis with support from all authors. The paper was written by R.L.J. with contribution from all authors.

## Competing interests

The authors declare no competing interests.
