## [Peer Review file · Nature Communications]

Antarctic glaciers export carbon-stabilised iron(II)-rich particles to the surface Southern Ocean

Corresponding Author: Dr Rhiannon Jones

Version 0:

Reviewer comments:

Reviewer #1

(Remarks to the Author)

This paper investigates the role of glaciers on the WAP in supplying particulate Fe into the coastal ocean. Analysis of a selection of particulate samples reveals Fe particles that are high in Fe(II) relative to total particulate Fe. The authors suggest that the relatively high proportion of Fe(II) in samples would make them more bioavailable. The authors observe a range of Fe(II) rich minerals and infer that because they exist in oxic waters at distance from their apparent source, then they must be stabilised. They show that the Fe(II) is often encased in a carbon matrix which the authors suggest should stabilise the Fe(II) and slow its oxidation into Fe(III), which is generally considered to be less bioavailable. The primary data set was collected from three sites in the WAP and includes detailed analysis of the particles using SXM, XANES and SEM-EDX techniques on approximately 30, 61 and 9 samples respectively. I think the combination of these techniques is quite powerful and provides interesting insights into the nature of the particles sampled. A second data set (LTER) is used to infer likely transport of the samples collected into off-shelf waters. The new data are interesting and valuable to the community. Overall, I found the article well written and logical. The data seems to be of high quality. I think addressing the following issues will help other readers understand and interpret the study more thoroughly.

1. Throughout the study, it is not always clear how many samples were used to support each of the claims. I suggest making this clearer in the text.
2. Excessive supplementary file. Some items seem unnecessary. For instance, why is supp table 2 needed when most of the data is shown in fig 2? Supp figure 6 and 7 don't seem to bring much to the article. Supp table 5 – is there a missing column heading? Supp table 6 could be included as text in supp note 1.
3. My main criticism is the lack of discussion on randomisation when targeting particles for SXM and XANES analysis. If “regions of interest” are targeted, then the ratios presented in figure 1 will be biased and not indicative of the general particle pool. This has implications for the conclusions that can be drawn from the analysis. If randomisation was performed, add this detail. If it was not, adjust the description in the discussion and conclusions accordingly.

Detailed comments:

84 – “Despite differing source geology” is stated without evidence presented or discussed. Please explain how this was determined and its expected impact on the results.

97 – please include the details of the statistical test that was run

101-102 – it is unclear if the “transport between stations” refers to the three stations that were sampled in 2020 or the series of stations that were sampled as part of the LTER 2015 data. Some of the stations at each study site for instance in Marian Cove are quite close (2.5km), so I think this statement needs justification perhaps using mean surface current flow data. The statement is likely justified, but it could be strengthened by providing more information.

108 – only 30 images from 61 samples were analysed. Why only a subset? How well does this subset of samples represent all of the particles collected? An indication of variability would strengthen this result. Particle samples can be highly heterogenous, so how was each area randomly selected for SXM analysis to avoid bias by picking out “hotspots” or large clumps of particles?

109 – “hotspots of Fe” implies that these areas were targeted. How did you avoid bias in sampling? This needs to be discussed in the MS. Likewise in the methods line 384, it states that ROI (regions of interest) were targeted. Is it possible that your study targeted particles that matched your interest in carbon and Fe co-location? If so, the relative proportions in Fig1 would not be representative of the particles in general as presented. I am not familiar with the image processing detailed in 391-395 so perhaps this has been dealt with, but your audience should not need to read another paper to ensure the data presented is representative of particles in general and therefore suitable to make generalisations such as line 40 of the abstract.

143 – add a comma after the first Fe

143 – some of your audience may take issue with using the phrase “rich in bioavailable Fe” when no bioassays have been performed in this study. Perhaps adjust terminology to reflect the inference from promising work in this area referenced in the introduction eg refs 6-10.

148 – line 84 mentioned differing source geology, and now you seem to be implying that the source geology is consistent across the WAP. Please clarify this apparent contradiction.

151 – “4.8% of particles scanned” – this statistic is not generally representative unless the subsampling was randomised. If so, be careful with extrapolations in conclusions.

Lines 160, 170 and 172 – some vague terminology could be strengthened with some statistics to support observations. “follows the Gade line”, “aligns with sediment inputs”, “paralleling with a drop in surface pFe” are all too vague in my opinion.

165 – this sentence lacks integration into the text and discussion of the importance of these concentrations in this context. Likewise, the final sentence of line 174.

180-184 – how can the authors rule out differences in the source rock geology when making this statement?

190 – “provide a large source” – it is unclear whether this statement is supported by the data set due to the lack of discussion on the randomisation of sampling. For instance, is the high relative proportion of Fe(II) only an artifact within the “regions of interest” or indicative of the sample set at large?

239 – “environmentally high” is strange terminology. Consider deleting “environmentally”.

241 – the term “labile” is often used in this field when a chemical leach (such as the Berger leach) has been completed, which was not done here. In this context it seems to be used interchangeably with bioavailable and on line 247 “reactive”, which may not be appropriate.

360 – state the number of samples as well as the number of stations.

Figure 2: the dots do not look pink on my screen. Suggest another color is chosen.

Figure 3 caption: there does not appear to be a negative correlation presented, so it is unclear why this is referenced in the final sentence of the caption.

Figure 4 – line 632 – I was very confused by the use of the term “contemporaneously” when I first read this caption as I thought it referred to the timing of the LTER data collection and the current data set! Just remove the term to avoid confusion.

Reviewer #2

(Remarks to the Author)

The manuscript by Jones et al entitled “Antarctic glaciers export carbon-stabilised iron(II)-rich particles to the surface Southern Ocean” by Jones et al. addresses the question of the link between melting Antarctic glaciers and bioavailable iron supply into the ocean. This question is currently little studied, and new data on the subject are particularly welcome due to the primary role of iron in controlling the biological carbon pump in the Southern ocean and the context of the accelerated melting of these glaciers under the effect of climate change.

For this process to have an influence on biogeochemistry and the carbon cycle in the Southern Ocean, 3 conditions must be met:

- 1) melting glaciers must be a significant source of iron
- 2) The form(s) of iron must be available to phytoplankton
- 3) The bioavailable form(s) of iron must be efficiently transported from the source to iron-limited areas of the Southern Ocean.

I have tried to assess below how the works presented in the manuscript provide new elements on these different aspects. On the first theme, the authors provide new data. Using a cutting-edge methodology, they demonstrate that small particles present in marine environments close to glaciers contain iron with valences II and III. These results are convincing. This analysis is completed by that of carbon, which shows that it is associated in a certain proportion with the same particle as that containing iron. I'm not a specialist in the method used, but in view of the spectra and the lack of detail they present, it's hard to believe that robust information on the composition of organic matter can be provided. I also note that the particles used for this analysis and presented in Figure 3 seem significantly larger (based on the scale bar of 1 μm) than most of those used to present the iron valence results (Figure 2) that are <1.6 μm . Why? What is the significance of these results on carbon for the smaller particles? The characterisation of the particles is completed by SEM-EDS. The resolution of this technic is around 1 μm so I wonder if the results presented in sup table 3 and sup figure 3 are here also related to the fraction of interest in the manuscript (<1.6 μm) or to larger particles. No EDS spectrum and no SEM images are provided. What is the protocol used to calibrate the instrument for EDS quantitative analysis? The authors mention line 147. “Scanning Electron Microscopy-Electron Dispersive 146 spectroscopy (SEM-EDS) reveals a primary aluminosilicate mineralogy, rich in Si, Al, Na, Ca, Mg and Fe, consistent with the geology of the WAP source rock [37-39] and indicative that these glaciogenic particles are predominantly lithogenic (Table S3).” But there is no data of, Na, Ca, Mg in the table 3 and linking aluminosilicate (a very broad category of minerals) to glaciogenic origin of the particles without further analysis is poorly convincing. The geochemical characterisation of the particles is incomplete. The origin of the particles (e.g. comparison with the composition of rocks at the different sites is not investigated in details). From a quantitative perspective there is no estimate of the flux of this fraction of small (FeII/FeIII) particles so it cannot be claimed that it is a significant source.

The theme 2) is related to the bioavailability of iron. The authors mention bioavailability 3 times in the abstract highlighting the importance of this issue. Contrary to previous studies (ref 9 and 11 in the manuscript) which have investigated the possible link between iron valence in particles and bioavailability using phytoplankton (strain or natural communities), the authors do not here provide new experimental data and simply rely on these published results to extrapolate that the percentage of iron Fe(II) found in their particles make them highly bioavailable for phytoplankton. This might be true or not, depending on a lot parameters, like the mineralogy, the age of the particles, the phytoplankton community composition... In

addition if the conclusion on the role of carbon to protect iron from oxidation is correct, carbon should also act as a barrier reducing the access to iron for phytoplankton. The consequence should be a decrease of the bioavailability. For the theme 3. The authors use oxygen isotope to track meteoritic water and seek for correlation with particulate trace metal. A similar data set was already published and discussed (Annet et al 2017). It does not help to prove that the small Fe(II)-containing particles discovered close to the glaciers are transported off shore. The concentration of particulate trace metal are determined for the entire range of size of particles and not only the smaller size. The use of the Stock's law is not very convincing because there are a lot of examples in the literature showing that this law does not correctly represents particles sinking in the marine environment for a lot of different reasons : physical (turbulence, mixing, shape of the particles...) and biogeochemical (aggregation/disaggregation due to biotic or abiotic processes). Overall I think that the conclusions of the study (line 239-242) "We show evidence that meltwaters sampled across the WAP export environmentally high concentrations of potentially bioavailable Fe(II)-rich particulates to Antarctic waters, and that a significant component of this pFe is aggregated with organic C. Export of this labile pFe by meltwaters beyond the nearshore environment and into potentially iron-limited waters is also demonstrated " are poorly supported by the data presented in the manuscript.

Reviewer #3

(Remarks to the Author)

Review of Jones et al. submission to Nature Communications

Upon reading this paper multiple times, I came away impressed with the approach and novelty of the coupled highly detailed analyses of particulate Fe and C chemistry using advanced techniques. The paper was well-written and is easy to read. It should definitely be published. My only issue is whether the really novel piece of the paper (in my estimation the particulate Fe(II)-C coupling component) is substantial and far reaching enough to warrant publication in Nature. I would argue that outside of establishing the Fe (II)-C coupling in glaciogenic particulate loads(Lines 94-140), the rest of the paper is making points that have made elsewhere by many others studying the composition and transport of glacial Fe fluxes to proximal ocean systems (in many of the papers cited by the authors!), albeit not for this particular region of Antarctica/Southern Ocean. To be clear, this is not a criticism of the quality of the work, but just something that the editor should consider in assessing whether the paper fits the journal-I could probably convince myself either way on this particular submission. Below I've added a few specific editorial comments that are hopefully useful, but I don't a ton of feedback because it was a pretty clean submission for which the authors should be commended.

Abstract-It takes too long to get to what you did here, and you really only highlight your Fe-C speciation results. Could you add some material about the more specific analysis and new conclusions that you came to when you integrated the other data (e.g. Mn, water isotopes, radium, etc). I would argue that the last two sentences are also already very well-established in the literature and have been for at least a decade. Highlighting the more specific conclusions drawn from your analysis here would help to highlight novelty.

Lines 94-140 Can you speculate on the provenance of the carbon based on the data that you collected, and if so, what does this say about how ubiquitous this mechanism may or may not be. Also, if the F(II) is 'entombed' and preserved by the organic matter in oxidizing conditions, you should then discuss the processes that would liberate the Fe to become bioavailable before offshore particle settling removes it from the mixed layer.

Lines 189-198 This paragraph needs to be referenced, as these points around fate and transport of particulate Fe have been demonstrated by many others.

Figure 2 legend. What is a 'lithogenic' standard relative to the other mineral samples that are color coded by valence? Were the others synthesized in the lab? I also wonder if you need so many standards on this plot that you don't reference in the text. Perhaps only include those that you discuss to clean up the plot a bit? It is a great plot for visualizing an impressive amount of suspended sediment synchrotron data.

In general, I thought the data visualizations were very strong in this submission.

Version 1:

Reviewer comments:

Reviewer #1

(Remarks to the Author)

Well done to the authors for considering the comments made by all the reviewers carefully. I think the authors have done a thorough job with the review and I have no further comments to make. Good luck finalizing the publication. Kind regards,
Pier

REVIEWER COMMENTS

Reviewer #1 (Remarks to the Author):

This paper investigates the role of glaciers on the WAP in supplying particulate Fe into the coastal ocean. Analysis of a selection of particulate samples reveals Fe particles that are high in Fe(II) relative to total particulate Fe. The authors suggest that the relatively high proportion of Fe(II) in samples would make them more bioavailable. The authors observe a range of Fe(II) rich minerals and infer that because they exist in oxic waters at distance from their apparent source, then they must be stabilised. They show that the Fe(II) is often encased in a carbon matrix which the authors suggest should stabilise the Fe(II) and slow its oxidation into Fe(III), which is generally considered to be less bioavailable. The primary data set was collected from three sites in the WAP and includes detailed analysis of the particles using SXM, XANES and SEM-EDX techniques on approximately 30, 61 and 9 samples respectively. I think the combination of these techniques is quite powerful and provides interesting insights into the nature of the particles sampled. A second data set (LTER) is used to infer likely transport of the samples collected into off-shelf waters. The new data are interesting and valuable to the community. Overall, I found the article well written and logical. The data seems to be of high quality. I think addressing the following issues will help other readers understand and interpret the study more thoroughly.

1. Throughout the study, it is not always clear how many samples were used to support each of the claims. I suggest making this clearer in the text.

Thank you for pointing this out. We have now added this information throughout the manuscript e.g. Lines 112-113, 153 – 159, 333, 352, 411, 469.

2. Excessive supplementary file. Some items seem unnecessary. For instance, why is supp table 2 needed when most of the data is shown in fig 2? Supp figure 6 and 7 don't seem to bring much to the article. Supp table 5 – is there a missing column heading? Supp table 6 could be included as text in supp note 1.

We thank the reviewer for this feedback. We think that supplementary table S2 is an important dataset that should be available for readers to use. The missing column heading in Table S5 is now addressed, with table note provided as well. Supplementary Figure 6 is often presented when using synchrotron techniques to show that the dwell times we use for STXM analysis do not cause beam damage of particles. We therefore think it is important to keep this information in to confirm this quality check was carried out and provide useful data on these kinds of samples to the community. Supplementary Figure 7 is used to illustrate the types of particles that should not sink below the 40 m mixed layer depth under the conditions described in the manuscript. We think this is a helpful visual

aid to the supplementary note, and our preference would be to retain it if possible, at the editors discretion. We have removed Table S6 and incorporated the relevant information into the text in supplementary note 1 and thank the reviewers for the feedback.

3. My main criticism is the lack of discussion on randomisation when targeting particles for SXM and XANES analysis. If "regions of interest" are targeted, then the ratios presented in figure 1 will be biased and not indicative of the general particle pool. This has implications for the conclusions that can be drawn from the analysis. If randomisation was performed, add this detail. If it was not, adjust the description in the discussion and conclusions accordingly.

Thank you for pointing this out- we have now addressed this in the detailed comment explanation below, referencing line 111 onwards, and methods lines 406 onwards.

Detailed comments:

84 – "Despite differing source geology" is stated without evidence presented or discussed. Please explain how this was determined and its expected impact on the results.

Apologies for this unintentional ambiguity, which we have changed to "provenance", reflecting that the particles are not sourced from the same place but from different islands along the WAP and therefore their consistency in Fe(II):Fe(III) ratio is somewhat surprising. We have also provided more detail on the regional geology further down (lines 152 - 168).

97 – please include the details of the statistical test that was run

We have now provided the t-test, with alpha stat, degrees of freedom, and p-value, starting on line 102.

101-102 – it is unclear if the "transport between stations" refers to the three stations that were sampled in 2020 or the series of stations that were sampled as part of the LTER 2015 data. Some of the stations at each study site for instance in Marian Cove are quite close (2.5km), so I think this statement needs justification perhaps using mean surface current flow data. The statement is likely justified, but it could be strengthened by providing more information.

We thank the reviewer for this suggestion. We refer to the three stations sampled in 2020, and have now made this clearer in the text. We have now looked at mean surface current flow data and provided references for each region, showing that taking the

maximum current speeds, the distance travelled would be less than the minimum distance, over 40-50 minutes (Line 102 - 109).

108 – only 30 images from 61 samples were analysed. Why only a subset? How well does this subset of samples represent all of the particles collected? An indication of variability would strengthen this result. Particle samples can be highly heterogenous, so how was each area randomly selected for SXM analysis to avoid bias by picking out “hotspots” or large clumps of particles?

We apologise for the lack of clarity here. 61 particles were imaged for the Fe and C XANES/NEXAFS, and 30 were imaged for the Fe and C difference maps.

For difference maps (e.g. those shown in Figure 2): our approach to particle selection was to use a low-resolution scan over a larger area (50 x 50 μm) to identify particles that have a transmission of between 20 – 80 %. Regions of interest must also have enough ‘white space’ i.e. background absorbance to facilitate a good L₀. Each chosen smaller region (usually ~5x5 μm) was then imaged at the carbon K-edge, and then the iron L-edge to produce the difference maps. This approach meant that at the time of selecting regions of interest, we had only an indication of particle presence and size, rather than knowledge of whether or not they contained either C or Fe. Again, due to the constraints of changing energies, we had to image all areas for C before we could image for Fe, so no bias could be introduced by confirming presence of C at each area (beam time is incredibly limited and it is essential to do all imaging at one energy at a time). Carbon was imaged first due to the potential damage to C by using the higher energy beam required for Fe. Critically, due to the study hypothesis, when analysing the data, Fe was considered first. There were 30 particles mapped for Fe presence that did contain Fe, and these were then considered for presence of C (19 contained carbon, 15 of which were significantly co-located). Indeed, Fe was essentially ubiquitous, and the co-presence of C with Fe is what we present in the manuscript. We have now made this clearer in the manuscript (L 406 onwards).

We used the following slightly different approach to measure the full Fe and C XANES/NEXAFS, using different, new regions: identify regions we get an Fe signal, move to the carbon edge and spectrally image the area, then move back up to the Fe edge and spectrally image the area. This was done to minimise any beam damage to carbon species (which is more beam sensitive than iron). Importantly, we found that iron was essentially ubiquitous in all our initial iron maps, and we rarely imaged a region that did not produce strong spectra/absorbance differences.

109 – “hotspots of Fe” implies that these areas were targeted. How did you avoid bias in sampling? This needs to be discussed in the MS. Likewise in the methods line 384, it states that ROI (regions of interest) were targeted. Is it possible that your study targeted particles that matched your interest in carbon and Fe co-location? If so, the relative

proportions in Fig1 would not be representative of the particles in general as presented. I am not familiar with the image processing detailed in 391-395 so perhaps this has been dealt with, but your audience should not need to read another paper to ensure the data presented is representative of particles in general and therefore suitable to make generalisations such as line 40 of the abstract.

We agree that particle selection could introduce bias into the results. We minimised any potential bias by using the method detailed above. More simply, particles were chosen based on the presence of detectable iron, with carbon NEXAFs taken after particles have been selected.

We have made this clearer in the results text from lines 406 onwards.

143 – add a comma after the first Fe *done*

143 – some of your audience may take issue with using the phrase “rich in bioavailable Fe” when no bioassays have been performed in this study. Perhaps adjust terminology to reflect the inference from promising work in this area referenced in the introduction eg refs 6-10.

Done, ‘potentially bioavailable’ is now used. At the request of the editor and other reviewer comments, we have also added a few additional references that bolster body of evidence in this area e.g. Fourquaz et al., 2023.

148 – line 84 mentioned differing source geology, and now you seem to be implying that the source geology is consistent across the WAP. Please clarify this apparent contradiction.

We have now provided additional information regarding the source geology across the WAP and the potential implications of this upon the samples (Lines 148 - 164). We have now used more recent references and discussion of these.

151 – “4.8% of particles scanned” – this statistic is not generally representative unless the subsampling was randomised. If so, be careful with extrapolations in conclusions.

This is a good point. The subsampling for SEM-EDS was indeed randomised, reflecting that only 4.8 % of particles measured had a very high (>20 wt%) Fe content. The bulk upper crust Fe wt% is ~3.4 wt%, and the particles on average comprised 4 - 8 Fe wt%, so those with a very high Fe wt% reflect Fe-enrichment relative to the continental crust average. It is notable that around 5% of the particles scanned had very high Fe content. Although this does not equate to a high dissolvable particulate component, the high pFe concentrations provided in the manuscript suggest that there is a significant amount of

digestible pFe present. We have clarified in the text that this high Fe wt% is consistent with XANES by identifying particles resembling both Fe(II) and Fe(III) minerals, such as Fe-bearing silicates and ferrihydrite (lines 162-168).

Lines 160, 170 and 172 – some vague terminology could be strengthened with some statistics to support observations. “follows the Gade line”, “aligns with sediment inputs”, “paralleling with a drop in surface pFe” are all too vague in my opinion.

We have reworded these sentences to include more specifics concerning the Gade line and details of the decline in optical transmission along it. The Gade line is a very well-established indicator of glacial meltwater, representing a mixing line (with known slope in potential temperature/salinity space) indicative of glacial ice melting in seawater. It is only ever segments of potential temperature/salinity curves that align with the Gade line, as the meltwater only impacts certain parts of the water column, thus correlations are not especially informative in this context. Instead, it is the similarity in slope of the part of the curve with lowered optical transmission that is instructive, demonstrating both the presence of glacial meltwater in that part of the water column and that the meltwater contributes to the particle load (as transmission is a proxy for suspended particulate matter). This approach (i.e. in the Figure S2 provided in the supplementary) follows that used in e.g. Meredith et al. Science Advances (2022) and many other previous works.

We avoided statistics for the drop in particulate metals with meteoric water at Borgen Bay as the total sample number for particulate metal concentrations in Borgen Bay is 3. (The r^2 is 0.99 – 1.00). However, we have clarified the writing to be more definitive on the point being made: that all metals decrease, as does the meteoric water component.

165 – this sentence lacks integration into the text and discussion of the importance of these concentrations in this context. Likewise, the final sentence of line 174.

These have now been integrated and discussed more, and linked with the sediment tracer data (see Lines 170 onwards).

180-184 – how can the authors rule out differences in the source rock geology when making this statement?

We have altered this discussion to include a comparison with SEM data, which is independent of total concentration. The SEM data shows that the median Al concentration in the particles measured was 6.5, 7.8, and 12.0 % for Anvers Island, Adelaide Island, and King George Island, respectively. Mean KGI Al wt% was therefore 1.5 - 1.75 times higher. The total digest data of particulate metals shows that digestible pAl at King George Island was 3 – 7 times higher than the SEM data. The SEM data is independent of sediment load, so comparing the two better indicates that KGI had a higher sediment load. We have

altered lines the text to include this, particularly in lines 200 – 207). The radium and thorium data published in Jones et al (2023) is also higher in the surface at KGI than Adelaide Island, which we reference to support this interpretation.

190 – “provide a large source” – it is unclear whether this statement is supported by the data set due to the lack of discussion on the randomisation of sampling. For instance, is the high relative proportion of Fe(II) only an artifact within the “regions of interest” or indicative of the sample set at large?

This is now explained better, and we include our approaches to randomisation in the text and above.

239 – “environmentally high” is strange terminology. Consider deleting “environmentally”.

Done.

241 – the term “labile” is often used in this field when a chemical leach (such as the Berger leach) has been completed, which was not done here. In this context it seems to be used interchangeably with bioavailable and on line 247 “reactive”, which may not be appropriate.

This is a fair comment and makes the text unclear so we have removed use of labile. We now use potentially bioavailable throughout, except on lines 133 – 137 when we are referring to types of organic C.

360 – state the number of samples as well as the number of stations.

This is now done.

Figure 2: the dots do not look pink on my screen. Suggest another color is chosen.

We enquire if the reviewer means Figure 1, as Fig. 2 has a legend (so colour should match legend) and the dots in Fig.2 are red.

We have put in the caption inf Fig.1 ‘red/pink’ and suggest to the editor that we see how the figure proofs look and we can change this accordingly.

If in this case the reviewer did mean Fig. 2, we avoided matching the ‘Fe valence’ boxes to the colour of the standard points because the valence boxes do not encompass all variability in measurements so we wish to avoid a strict definition of where a standard would sit in time and space.

Figure 3 caption: there does not appear to be a negative correlation presented, so it is unclear why this is referenced in the final sentence of the caption.

This has now been removed from the caption to avoid confusion.

Figure 4 – line 632 – I was very confused by the use of the term “contemporaneously” when I first read this caption as I thought it referred to the timing of the LTER data collection and the current data set! Just remove the term to avoid confusion.

Done.

Reviewer #2 (Remarks to the Author):

The manuscript by Jones et al entitled “Antarctic glaciers export carbon-stabilised iron(II)-rich particles to the surface Southern 1Ocean” by Johnes et al. addresses the question of the link between melting Antarctic glaciers and bioavailable iron supply into the ocean. This question is currently little studied, and new data on the subject are particularly welcome due to the primary role of iron in controlling the biological carbon pump in the Southern ocean and the context of the accelerated melting of these glaciers under the effect of climate change.

For this process to have an influence on biogeochemistry and the carbon cycle in the Southern Ocean, 3 conditions must be met:

- 1) melting glaciers must be a significant source of iron
- 2) The form(s) of iron must be available to phytoplankton
- 3) The bioavailable form(s) of iron must be efficiently transported from the source to iron-limited areas of the Southern Ocean.

I have tried to assess below how the works presented in the manuscript provide new elements on these different aspects.

On the first theme, the authors provide new data. Using a cutting-edge methodology, they demonstrate that small particles present in marine environments close to glaciers contain iron with valences II and III. These results are convincing. This analysis is completed by that of carbon, which shows that it is associated in a certain proportion with the same particle as that containing iron. I'm not a specialist in the method used, but in view of the spectra and the lack of detail they present, it's hard to believe that robust information on the composition of organic matter can be provided. I also note that the particles used for this analysis and presented in Figure 3 seem significantly larger (based on the scale bar of 1 μm) than most of those used to present the iron valence results (Figure 2) that are $< 1.6 \mu\text{m}$. Why? What is the significance of these results on carbon for the smaller particles?

The images in Figure 2 were provided as clean examples for visualisation i.e. the reviewer's interpretation above is based on these images, and not our full dataset. At the smaller particle scale, regions still correlated with Fe and C and provided clean spectra but were less good for visualisation (i.e. pixelated). Figure S1 shows that several small particles are often present in an area of interest (within <microns of each other), and that these particles often produce different XANES. Our analysis and interpretations are based on the full set of spectra from all particles, with full data in supplementary/online, meaning our conclusions are drawn robustly across all sizes.

Given the complex nature of natural organic matter (which can consist of thousands of different compounds), carbon NEXAFS are difficult to quantitatively interpret (e.g. via full peak deconvolution), so we have chosen not to do so. However, C NEXAFS using STXM is a sensitive technique for distinguishing C speciation and bonding in organic molecules, particularly the key functional groups present (the identification of which depends on the energy at which we can see distinguishable peaks/can deconvolute peaks in spectra). This makes C NEXAFS useful for exploring the key functional groups associated with particulate organic matter - it has been used like this across a number of Earth Science disciplines (Barber et al., 2024; Lalonde et al., 2012), including to investigate the C composition of marine particles (Hoffman et al., 2018; Toner et al., 2009; Moore et al., 2023).

We agree with the reviewer that interpretation of C spectra is not straightforward, but we feel this is fully disclosed in the manuscript given our statement about the difficulties of assigning peaks in the spectra to specific carbon functional groups in the presence of e.g. oxyhydroxides. We have noted the main groups that are suggested by our data (drawing on our expertise e.g. Moore et al., 2023 (Nature); Curti et al., 2021 (Nature Earth and Environment), and note in the manuscript that more research on natural samples is needed.

The characterisation of the particles is completed by SEM-EDS. The resolution of this technic is around 1 μm so I wonder if the results presented in sup table 3 and sup figure 3 are here also related to the fraction of interest in the manuscript (<1.6 μm) or to larger particles. No EDS spectrum and no SEM images are provided. What is the protocol used to calibrate the instrument for EDS quantitative analysis?

We offer the SEM-EDS data as characterisation of the mineralogy of particles found in the surface ocean, but use the particulate metals data to characterise the digestible/leachable component of particles i.e. that which would be important for bioavailability of particles. We have added a substantial contribution to the SEM-EDS section of the manuscript to improve the use and relevance of this dataset, which we discuss in more detail below (see lines 148 onwards).

Quantification of element abundance was calibrated against the suite of SEM-EDS AZtec in-built standards, which necessitates all analyses being normalised to 100%, and elements of interest to have an atomic number > 11, of which both conditions were met. The inbuilt AZtec EDS calibration setup requires routine calibration of the beam energy and the beam resolution, which was performed using a cobalt metal standard at 20 kV and a working distance of 15 mm. Routine calibration of the beam energy and beam resolution was performed using a cobalt metal standard at 20 kV and a working distance of 15 mm. We have now provided this information in the manuscript. We have also now included some spectra in the supplementary (but to avoid a congested supplementary kept this to one from each bay). We also provide a DOI for the spectra in the data availability.

The authors mention line 147. "Scanning Electron Microscopy-Electron Dispersive 146 spectroscopy (SEM-EDS) reveals a primary aluminosilicate mineralogy, rich in Si, Al, Na, Ca, Mg and Fe, consistent with the geology of the WAP source rock [37-39] and indicative that these glaciogenic particles are predominantly lithogenic (Table S3)." But there is no data of, Na, Ca, Mg in the table 3 and linking aluminosilicate (a very broad category of minerals) to glaciogenic origin of the particles without further analysis is poorly convincing.

The geochemical characterisation of the particles is incomplete. The origin of the particles (e.g. comparison with the composition of rocks at the different sites is not investigated in details).

We thank the reviewer for this feedback, and have rewritten the geochemical characterisation section in the text with additional important parameters, and now include additional information on the geology of the individual regions in the paper with updated references (Lines 148 onwards). We have also provided the quantified component of Mg, Ca and Na in the Table S2 and provide the full SEM wt% in the online dataset.

The interpretation that the particles are glaciogenic in origin is predicated on the full set of datasets presented in the paper (stable oxygen isotope and sediment tracers, proximity to glaciers, surface ocean sampling, total element concentrations), rather than the mineralogy alone. We have therefore altered the point of introduction of the SEM-EDS analysis to after the discussion of each site characteristics, to highlight that it is not the central dataset. Our interpretation is that minerals more authigenic in nature would be expected if the particles were not derived directly from glacier meltwater but instead were a) from diagenetic sediments or b) atmospherically deposited. However, it may not have been fully clear that the mineralogy fits with the interpretation rather than drives the interpretation. That Fe(II) bearing silicates are found in glacial particles, and that these are more bioavailable than Fe(III) silicates and oxyhydroxides, has been demonstrated in many studies on particles of glaciogenic origin (Raiswell et al., (2008); Raiswell et al., (2010); Shoenfelt et al., (2017); Hawkings et al., (2018)), and the high reactive component of Fe(II)-

bearing silicates is further demonstrated in Raiswell et al., (2010) and Raiswell et al., (2018) and references therein.

From a quantitative perspective there is no estimate of the flux of this fraction of small (FeII/FeIII) particles so it cannot be claimed that it is a significant source.

The particles we identified using STXM had ~40% Fe(II) content on average. We would argue that the randomised approach to sampling makes it likely that our data are broadly representative of the particle pool at large, and the dispersed particles identified (~<micron) are likely to stay in suspension. We find that the Fe(II) content is high, and the particulate Fe concentrations are environmentally very high, so suggest that this is a reasonable interpretation with the data that we generated. Work analysing the relative contribution of particulate and nanoparticulate fractions (and then subsequent STXM analysis) would be very interesting but is a very large study beyond the scope of this (and dependent on many things, including securing a significant amount successful synchrotron time, i.e. months!).

These are the first data to quantify the content of Fe(II) vs Fe(III) in glacial particles in the Antarctic marine environment, and given the paradigm that most Fe will be present as Fe(III) in an oxic ocean, finding such a high proportion of particulate Fe(II) is very exciting in light of the studies that find this particulate form more bioavailable.

The theme 2) is related to the bioavailability of iron. The authors mention bioavailability 3 times in the abstract highlighting the importance of this issue. Contrary to previous studies (ref 9 and 11 in the manuscript) which have investigated the possible link between iron valence in particles and bioavailability using phytoplankton (strain or natural communities), the authors do not here provide new experimental data and simply rely on these published results to extrapolate that the percentage of iron Fe(II) found in their particles make them highly bioavailable for phytoplankton. This might be true or not, depending on a lot parameters, like the mineralogy, the age of the particles, the phytoplankton community composition... In addition if the conclusion on the role of carbon to protect iron from oxidation is correct, carbon should also act as a barrier reducing the access to iron for phytoplankton. The consequence should be a decrease of the bioavailability.

We agree with the reviewer that the current body of literature on the bioavailability of Fe(II) particles is not complete and that further research is required. There is evidence that the most reactive forms of particulate Fe such as ferrihydrite are also somewhat bioavailable (based on incubation with phytoplankton cultures) (e.g. Shoenfelt et al., 2017), and, critically, an exciting recent paper that demonstrated that Fe was more bioavailable to phytoplankton near to glaciers, where particulate Fe(II) is prevalent (Fourquaz et al. 2023). Our research adds to the body of evidence that suggests glaciogenic material is either

Fe(II)-rich or mixed valence, and therefore has a higher bioavailability potential than aged iron oxides as more prevalent in e.g. atmospheric dust.

We do however concede that there are gaps in the literature on the topic, and there could be some potential to mislead the reader. We have revised our terminology to 'potentially bioavailable' (e.g. lines 34, 217, 271, 275) so as not to confuse readers, since we have not directly measured it. We have also rephrased the abstract introduction to this as 'understood to be more bioavailable'. We hope that the above reassures the reviewer and thank them for bringing this to our attention.

The C NEXAFS indicates that its composition is rich in carboxylic functional groups, not more refractory aromatic functional groups, often associated with labile material. This has been observed elsewhere (Toner et al. 2009). However, we cannot tell whether this makes the Fe more or less labile. The emphasis that we make is therefore that glaciogenic material represents a relatively significant potentially bioavailable particulate component in comparison to other particulate sources.

For the theme 3. The authors use oxygen isotope to track meteoritic water and seek for correlation with particulate trace metal. A similar data set was already published and discussed (Annet et al 2017). It does not help to prove that the small Fe(II)-containing particles discovered close to the glaciers are transported off shore. The concentration of particulate trace metal are determined for the entire range of size of particles and not only the smaller size.

Although the reviewer has a good point, we would argue that smaller particles are much more likely to remain suspended, and therefore, transported offshore. Nanoparticulate aggregates will be impacted by aggregation/disaggregation, biotic and abiotic processes, as well as mixing. The 2015 LTER dataset is important, because it uses a well-established stable oxygen isotope technique to trace meltwater transport across the WAP (e.g. Moffat and Meredith, 2018; Meredith et al., 2018; Meredith et al., 2013.), enabling correlations with a substantial across-shelf particulate metal dataset. Correlating this with a substantial across-shelf particulate metal dataset, we think provides a compelling case that these high metal concentrations are associated with glacial meltwater transit. We believe that the inclusion of the pMn data showing the linear relationship with meteoric water (and distance offshore) further bolsters the interpretation that the pFe is linked with meteoric water movement, and not with a presumably large yet unidentified source. Of course, we have also considered the potential for other sources and find no compelling evidence of these – dust deposition in the region is extremely low, and shelf sediments (although another important part of the Southern Ocean Fe cycle) would presumably have quite a different profile, requiring offshore upwelling to the surface ocean (2 – 5 m depth).

The use of the Stokes' law is not very convincing because there are a lot of examples in the literature showing that this law does not correctly represents particles sinking in the marine environment for a lot of different reasons : physical (turbulence, mixing, shape of the particles...) and biogeochemical (aggregation/disaggregation due to biotic or abiotic processes).

We agree that Stokes' Law is a simplification of the processes at play but we would argue that this, in addition to evidence from oxygen isotopes and pFe, provides a convincing argument that particles do reach the self. Therefore, an idealised approximation of the sinking rate is appropriate to interpret the concentration data. We have now made it clear in the text that this is a first-order approximation (Line 264) appropriate for the discussion, and do already discuss the limitations associated with our interpretations in the supplementary. Perhaps also of interest here is a Nature journal study that suggests glaciogenic material forms quite buoyant flocs (Markussen et al., 2016) i.e. we do not know the net effect of additional mechanisms, and offer a large potential range of values for particle density in our idealised model.

Overall I think that the conclusions of the study (line 239-242) "We show evidence that meltwaters sampled across the WAP export environmentally high concentrations of potentially bioavailable Fe(II)-rich particulates to Antarctic waters, and that a significant component of this pFe is aggregated with organic C.

Export of this labile pFe by meltwaters beyond the nearshore environment and into potentially iron-limited waters is also demonstrated " are poorly supported by the data presented in the manuscript.

We respectfully disagree with this comment, and hope that we have addressed the reviewers concerns in our response above and in the manuscript. Our conclusions are based on the following:

- *The concentrations of pFe are environmentally high in the context of the marine environment, i.e. in the 30 – 300 nM range proximal to the glacier, and in the 0.1 – 10 nM range in the coastal range, compared to <0.1 nM, typical of the southern Ocean.*
- *Demonstration of a statistically significant level of both high Fe(II) content (41 +/- 8 %, n = 61) and Fe association with C.*
- *Oxygen isotope data both demonstrating that high particulate and dissolved metals are found in high meltwater environments (i.e. proximal to the glacier), and showing that meltwater and particulate material is likely transported over the shelf*

Our conclusions are based on a combination of well-established and relatively new analytical techniques, providing convincing and novel findings. To complement our

findings and discuss the implications, we use the findings of others to highlight the likely lability of the particulate material.

We are grateful to the reviewer for their feedback, which has helped us provide a stronger manuscript, and hope that the adaptations we have made to the text show this.

Reviewer #3 (Remarks to the Author):

Review of Jones et al. submission to Nature Communications

Upon reading this paper multiple times, I came away impressed with the approach and novelty of the coupled highly detailed analyses of particulate Fe and C chemistry using advanced techniques. The paper was well-written and is easy to read. It should definitely be published. My only issue is whether the really novel piece of the paper (in my estimation the particulate Fe(II)-C coupling component) is substantial and far reaching enough to warrant publication in Nature. I would argue that outside of establishing the Fe (II)-C coupling in glaciogenic particulate loads(Lines 94-140), the rest of the paper is making points that have made elsewhere by many others studying the composition and transport of glacial Fe fluxes to proximal ocean systems (in many of the papers cited by the authors!), albeit not for this particular region of Antarctica/Southern Ocean. To be clear, this is not a criticism of the quality of the work, but just something that the editor should consider in assessing whether the paper fits the journal-I could probably convince myself either way on this particular submission. Below I've added a few specific editorial comments that are hopefully useful, but I don't a ton of feedback because it was a pretty clean submission for which the authors should be commended.

Abstract-It takes too long to get to what you did here, and you really only highlight your Fe-C speciation results. Could you add some material about the more specific analysis and new conclusions that you came to when you integrated the other data (e.g. Mn, water isotopes, radium, etc). I would argue that the last two sentences are also already very well-established in the literature and have been for at least a decade. Highlighting the more specific conclusions drawn from your analysis here would help to highlight novelty.

Thank you for this insightful feedback. We have taken this onboard and adjusted the abstract to also highlight the additional data. We have shortened the first half of the abstract and therefore retained the last two sentences as we would like these to link our findings with the bigger picture.

Lines 94-140 Can you speculate on the provenance of the carbon based on the data that

you collected, and if so, what does this say about how ubiquitous this mechanism may or may not be.

We have provided speculation on this on lines 133 - 146, and believe that these sorts of processes are likely commonplace in the marine organic matter pool – which is also demonstrated elsewhere in the literature. We suggest that the source of organic C will be microbial in origin, because of the lack of vegetative inputs in the region. In terms of direct provenance this is difficult without endmember sources, but we hope that the analysis provided elucidates the likely pathways.

Also, if the Fe(II) is 'entombed' and preserved by the organic matter in oxidizing conditions, you should then discuss the processes that would liberate the Fe to become bioavailable before offshore particle settling removes it from the mixed layer.

We thank the reviewer for this comment. Lines 138 – 141 now discuss the potential bioavailability of labile organic matter-bound Fe.

Lines 189-198 This paragraph needs to be referenced, as these points around fate and transport of particulate Fe have been demonstrated by many others.

We have now added in three important references and a little more text.

Figure 2 legend. What is a 'lithogenic' standard relative to the other mineral samples that are color coded by valence? Were the others synthesized in the lab? I also wonder if you need so many standards on this plot that you don't reference in the text. Perhaps only include those that you discuss to clean up the plot a bit? It is a great plot for visualizing an impressive amount of suspended sediment synchrotron data.

Apologies that this is unclear. We have cleaned the plot up, by removing some of the standards that aren't discussed, and/or were duplicated standards, and by removing some of the annotation text that refers to standards we don't reference. To demonstrate the range of Fe(III) to Fe(II) in STXM space, we suggest that we retain those in the new figure (i.e. the Fe(II) standards that are not specifically discussed), and keep the labels that are needed/don't cluster too much.

Regarding the lithogenic v.s. Fe(II) this was an issue with the data file to MATLAB plotting code that we hadn't noticed. Sorry about that – lithogenic and Fe(II) standards were equivalent. We've now grouped them all in Fe(II).

In general, I thought the data visualizations were very strong in this submission.

We thank the reviewer for this comment!

References:

- Barber, A., Mirzaei, Y., Brandes, J., Joshani, A., Gobeil, C., & G  linas, Y. (2024). Redox conditions influence the chemical composition of iron-associated organic carbon in boreal lake sediments: A synchrotron-based NEXAFS study. *Geochimica et Cosmochimica Acta*, 382, 51-60.
- Curti, L., Moore, O. W., Babakhani, P., Xiao, K. Q., Woulds, C., Bray, A. W., ... & Peacock, C. L. (2021). Carboxyl-richness controls organic carbon preservation during coprecipitation with iron (oxyhydr) oxides in the natural environment. *Communications Earth & Environment*, 2(1), 229.
- Fourquez, M., Janssen, D. J., Conway, T. M., Cabanes, D., Ellwood, M. J., Sieber, M., ... & Hassler, C. (2023). Chasing iron bioavailability in the Southern Ocean: Insights from *Phaeocystis antarctica* and iron speciation. *Science Advances*, 9(26), eadf9696.
- Hoffman, C. L., Nicholas, S. L., Ohnemus, D. C., Fitzsimmons, J. N., Sherrell, R. M., German, C. R., ... & Toner, B. M. (2018). Near-field iron and carbon chemistry of non-buoyant hydrothermal plume particles, Southern East Pacific Rise 15 S. *Marine Chemistry*, 201, 183-197.
- Lalonde, K., Mucci, A., Ouellet, A., & G  linas, Y. (2012). Preservation of organic matter in sediments promoted by iron. *Nature*, 483(7388), 198-200.
- Markussen, T. N., Elberling, B., Winter, C., & Andersen, T. J. (2016). Flocculated meltwater particles control Arctic land-sea fluxes of labile iron. *Scientific reports*, 6(1), 24033.
- Meredith, M. P., Venables, H. J., Clarke, A., Ducklow, H. W., Erickson, M., Leng, M. J., ... & van den Broeke, M. R. (2013). The freshwater system west of the Antarctic Peninsula: spatial and temporal changes. *Journal of Climate*, 26(5), 1669-1684.
- Meredith, M. P., Falk, U., Bers, A. V., Mackensen, A., Schloss, I. R., Ruiz Barlett, E., ... & Abele, D. (2018). Anatomy of a glacial meltwater discharge event in an Antarctic cove. *Philosophical Transactions of the Royal Society A: Mathematical, Physical and Engineering Sciences*, 376(2122), 20170163.
- Moffat, C., & Meredith, M. (2018). Shelf-ocean exchange and hydrography west of the Antarctic Peninsula: a review. *Philosophical Transactions of the Royal Society A: Mathematical, Physical and Engineering Sciences*, 376(2122), 20170164.
- Moore, O. W., Curti, L., Woulds, C., Bradley, J. A., Babakhani, P., Mills, B. J., ... & Peacock, C. L. (2023). Long-term organic carbon preservation enhanced by iron and manganese. *Nature*, 621(7978), 312-317.

Raiswell, R., Hawkings, J., Elsenousy, A., Death, R., Tranter, M., & Wadham, J. (2018). Iron in glacial systems: Speciation, reactivity, freezing behavior, and alteration during transport. *Frontiers in Earth Science*, 6, 222.

Raiswell, R., Benning, L. G., Tranter, M., & Tulaczyk, S. (2008). Bioavailable iron in the Southern Ocean: the significance of the iceberg conveyor belt. *Geochemical transactions*, 9, 1-9.

Raiswell, R., Vu, H. P., Brinza, L., & Benning, L. G. (2010). The determination of labile Fe in ferrihydrite by ascorbic acid extraction: methodology, dissolution kinetics and loss of solubility with age and de-watering. *Chemical Geology*, 278(1-2), 70-79.

Shoenfelt, E. M., Sun, J., Winckler, G., Kaplan, M. R., Borunda, A. L., Farrell, K. R., ... & Bostick, B. C. (2017). High particulate iron (II) content in glacially sourced dusts enhances productivity of a model diatom. *Science advances*, 3(6), e1700314.

Shoenfelt, E. M., Winckler, G., Annett, A. L., Hendry, K. R., & Bostick, B. C. (2019). Physical weathering intensity controls bioavailable primary iron (II) silicate content in major global dust sources. *Geophysical Research Letters*, 46(19), 10854-10864.

Toner, B. M., Fakra, S. C., Manganini, S. J., Santelli, C. M., Marcus, M. A., Moffett, J. W., ... & Edwards, K. J. (2009). Preservation of iron (II) by carbon-rich matrices in a hydrothermal plume. *Nature Geoscience*, 2(3), 197-201.

Wyatt, N. J., Birchill, A., Ussher, S., Milne, A., Bouman, H. A., Shoenfelt Troein, E., ... & Moore, C. M. (2023). Phytoplankton responses to dust addition in the Fe–Mn co-limited eastern Pacific sub-Antarctic differ by source region. *Proceedings of the National Academy of Sciences*, 120(28), e2220111120.